# Long-term outcomes after discontinuing biological drugs and tofacitinib in patients with rheumatoid arthritis: A prospective cohort study

**Shunsuke Mori**[1]*, **Akitomo Okada**[2], **Tomohiro Koga**[3,4], **Yukitaka Ueki**[5]

1 Department of Rheumatology, Clinical Research Center for Rheumatic Diseases, National Hospital Organization Kumamoto Saishun Medical Center, Kohshi, Kumamoto, Japan, 2 Department of Rheumatology, Japanese Red Cross Nagasaki Genbaku Hospital, Nagasaki, Japan, 3 Division of Advanced Preventive Medical Sciences, Department of Immunology and Rheumatology, Nagasaki University Graduate School of Biomedical Sciences, Nagasaki, Japan, 4 Center for Bioinformatics and Molecular Medicine, Nagasaki University Graduate School of Biomedical Sciences, Nagasaki, Japan, 5 Rheumatic and Collagen Disease Center, Sasebo Chuo Hospital, Sasebo, Nagasaki, Japan

* mori.shunsuke.ra@mail.hosp.go.jp

# Abstract

## Objective

This study examined long-term outcomes of biological disease-modifying antirheumatic drugs (bDMARDs) and tofacitinib discontinuation in patients with rheumatoid arthritis (RA).

## Methods

Ninety-seven RA patients who desired drug discontinuation after sustained remission or low disease activity for at least 48 weeks due to stable treatment with biological drugs or tofacitinib were enrolled into this study. All patients were prospectively followed until disease flare or the end of the study. Discontinued drugs (previous drugs) were reintroduced to treat flares.

## Results

Following bDMARD/tofacitinib discontinuation (mean follow-up, 2.1 years; standard deviation, 2.0), disease flare occurred at a crude incidence rate of 0.36 per person-year. The median time to flare was 1.6 years (95% confidence interval [CI] 0.9–2.6), and the cumulative flare probability was estimated to be 45% at 1 year, 64% at 3 years, and 80% at 5 years. No or little radiological progression was shown in 87.1% of patients who maintained remission for 3 years. A Fine–Gray competing risk regression analysis showed that predictive factors for a flare were longer RA duration at the start of bDMARD/tofacitinib treatment, previous failure of treatment with bDMARDs, and a shorter period of remission or low disease activity before drug discontinuation. Type of discontinued drug was not identified as a predictive factor after adjusting for other predictor variables. Restarting previous treatment regimens led to rapidly regaining disease control in 89% of flare patients within 1 month.

**Data Availability Statement:** All data supporting the findings are available from the Human Research Ethics Committee of National Hospital

Organization Kumamoto Saishun Medical Center for all interested researchers who meet the criteria for access to confidential data. Because these data include patients' personal information, the Committee does not recommend that such data be made public unnecessarily. Please contact Mr. Hiroshi Tsumura, the Control Manager of the Committee, at 616-kikaku@mail.hosp.go.jp to request the data.

**Funding:** This study was supported by research funds from the National Hospital Organization, Japan. The funder had no role in the study design, data collection and analysis, decision to publish, or manuscript preparation.

**Competing interests:** S. Mori has received lecture fees from Pfizer Japan Inc., Eli Lilly Japan K.K., and Asahikasei Pharma. The other authors have declared that no conflicts of interest exist. This does not alter our adherence to PLOS ONE policies on sharing data and materials.

## Conclusion

Discontinuation of bDMARD/tofacitinib may be a feasible strategy in RA patients, especially patients with early treated and longer-controlled RA. Flares are manageable in most RA patients and radiological progression is rare for at least 3 years in patients with sustained remission after bDMARD/tofacitinib discontinuation.

## Introduction

Rheumatoid arthritis (RA) is a chronic immune-mediated inflammatory disease that causes progressive damage to joint cartilage and bone, disability, and comorbidity [1]. RA has become a controllable disease. Long-term health-related patient quality of life has dramatically improved because of the availability of conventional, biological, and new non-biological targeted disease-modifying antirheumatic drugs (DMARDs) as well as application of early aggressive intervention and the treat-to-target approach with tight control of disease activity [2]. Today, clinical remission early in the disease course, with a low disease activity (LDA) as a best feasible alternative, is the therapeutic target for every RA patient. Maintaining a state of remission or LDA will relieve clinical signs and symptoms, inhibit the occurrence or progression of structural damage, and improve or normalize physical function and social and work-related activities [3–5]. However, adverse events and high healthcare costs associated with life-long immune-modulatory treatment are significant burdens on patients as well as rheumatologists. Patients frequently express a desire to decrease the dose and for drug holidays [6, 7].

Many studies have addressed the impact of tapering and/or discontinuation of biological DMARDs (bDMARDs) on clinical outcomes in RA patients who have achieved and maintained remission or LDA, including prospective observational studies, randomized controlled trials (RCTs), and open-label extension studies [8–12]. Outcomes of dose reduction and/or discontinuation of targeted synthetic DMARDs (tsDMARDs) have also been reported [13–15]. These studies suggested that a subset of patients may maintain remission or LDA after tapering and even discontinuation of bDMARDs and tsDMARDs. In most studies, however, follow-up periods after tapering and discontinuation of drugs were within 1 year.

In this observational study, we examined the long-term outcomes of bDMARD and tsDMARD discontinuation in RA patients who had maintained remission or LDA for ≥48 weeks. The cumulative probability of disease flare, predictive factors, radiological progression rate, and rescue therapy results were examined.

## Patients and methods

### Patients

Participants in this study were patients with RA who desired drug discontinuation after maintaining remission or LDA for ≥48 weeks through stable drug treatment with bDMARDs or tsDMARDs from January 2007 to June 2020 in the Rheumatology department of the National Hospital Organization Kumamoto Saishun Medical Center. All participants were over 18 years of age at enrollment and fulfilled the 1987 American College of Rheumatology (ACR) criteria or the 2010 ACR/European League Against Rheumatism (EULAR) criteria for diagnosis of RA [16, 17]. Participants were also required to have had a high or moderate disease activity at the start of bDMARD/tsDMARD treatment despite previous methotrexate (MTX) monotherapy for ≥3 months. Additionally, the patients were required to have already tapered off prednisolone at the start of bDMARD/tsDMARD discontinuation.

## Discontinuation of bDMARDs and tsDMARDs

All participants started the bDMARD and tsDMARD discontinuation after maintaining remission or LDA for ≥48 weeks, and they were prospectively followed-up to assess disease activity. A direct discontinuation strategy without stepwise tapering was implemented. All patients were recommended to return for a follow-up visit every 8 weeks after bDMARD/tsDMARD discontinuation. They were allowed to receive a stable dose of MTX (6 to 12 mg/week) during follow-up. They were not allowed to use other conventional synthetic DMARDs (csDMARDs) or prednisolone to control disease activity during follow-up. Follow-up started on the day of bDMARD/tsDMARD discontinuation and ended at the earliest date of the following: flare occurrence, lost to follow-up, death, adverse events, or the last follow-up visit before July 1, 2021. Loss to follow-up was defined as missing at least two scheduled visits without any contact. Adverse events were defined as those that caused MTX discontinuation. Decisions to discontinue MTX due to adverse events were made by the treating physicians on the basis of a comprehensive evaluation of physical findings, laboratory findings, and radiological examinations at each visit or at any time during follow-up if patients had clinical signs or symptoms.

## Rescue therapy

Patients who experienced a disease flare started rescue therapy, which comprised restarting the discontinued drugs (previous drugs). Disease activity was examined every 2 weeks during rescue therapy. When disease activity was not controlled by the previous drugs within 4 weeks, patients were treated with another bDMARD or tsDMARD. The choice of the other bDMARD or tsDMARD therapy was at the discretion of each treating physician.

## DMARDs

The bDMARDs used in this study included the following two classes: tumor necrosis factor (TNF) inhibitors (infliximab, etanercept, adalimumab, golimumab, and certolizumab pegol) and an interleukin-6 inhibitor (tocilizumab). Tocilizumab was administered via an intravenous infusion at 8 mg/kg every 4 weeks or by a subcutaneous injection of 162 mg every other week. For patients who failed to achieve remission or LDA by the original subcutaneous tocilizumab regimen at 3 months, a 162-mg once-weekly regimen was used. In this study, tofacitinib was the only tsDMARD used to treat the participants. Tofacitinib was administered at a dose of 5 mg twice a day.

## RA disease activity

The clinical disease activity index (CDAI) was used to quantify RA disease activity. Cut-off values for disease activity states were defined as follows: high disease activity, CDAI >22; moderate disease activity, CDAI >10 and ≤22; LDA, CDAI >2.8 and ≤10; and remission, CDAI ≤2.8 [18, 19]. After bDMARD/tsDMARD discontinuation, CDAI assessments were performed at each scheduled visit (usually every 8 weeks) or at any time if patients had clinical signs or symptoms. A disease flare was defined as a worsening of CDAI values (a return to moderate or high CDAI) at any time during follow-up.

## Assessment of joint destruction

Radiographs of the hands and feet were taken at the start of treatment with bDMARD or tsDMARD, at the start of bDMARD/tsDMARD discontinuation, and every year during follow-up. These data were used to examine structural outcomes after bDMARD/tsDMARD

discontinuation. Each radiograph was assessed independently by two readers (AO and TK) who were well trained and competent to score radiographs using the van der Heijde-modified total Sharp score (mTSS) [20]. The readers were also blinded to the patients' clinical status and treatment and to the chronologic order of the films. To ensure objectivity of the scoring, the average of the mTSS scores determined by each of the two readers was used for each patient's mTSS. A change in mTSS between the bDMARD/tsDMARD discontinuation and 3 years later (ΔmTSS) was determined for each patient who had maintained remission for 3 years or more. The proportion of patients with no (ΔmTSS ≤0 per 3 years) and little (ΔmTSS ≤1.0 per 3 years) radiological progression, which was defined as radiographic remission in this study, was determined [20].

## Ethical approval

This study was conducted in accordance with the principles of the Declaration of Helsinki. The protocol of this study also met the requirements of the Ethical Guidelines for Medical and Health Research Involving Human Subjects, Japan and was approved by the Human Research Ethics Committees of National Hospital Organization Kumamoto Saishun Medical Center (Nos. 31–36 and 2–12). Informed written consent was obtained from all participants before enrollment into this study.

## Statistical analysis

Mean and standard deviation (SD) were used as descriptive statistics for data with a continuous distribution, which included non-normally distributed data [21, 22], and as the number (percentage) for categorical data. Crude incidence rates (IRs) of flare and the 95% confidence intervals (CIs) were also calculated by dividing the number of incidence cases by the number of corresponding follow-up person-years (PYs).

The probability of a disease flare event occurring over time after drug discontinuation was estimated using the cumulative incidence function (CIF) because we considered the presence of competing risks in the present study (i.e., lost to follow-up). The occurrence of a competing risk event precludes the occurrence of the primary event of interest. In the presence of competing risks, the simple use of the Kaplan–Meier survival function can overestimate the cumulative incidence probability of a flare event. To avoid this possibility, we used the CIF instead of the Kaplan–Meier survival function. The median time to disease flare (95% CI) and the cumulative flare probability (95% CI) were estimated using the CIF. Gray's test with the *post hoc* Holm's procedure was used to test the equality of CIF plots among the previous drugs. Gray's test is the analogue to the log-rank test that is used for testing the equality of Kaplan–Meier survival curves between groups [23, 24].

Considering the presence of competing risks, Fine–Gray competing risk regression analysis was used to evaluate the effect of clinical characteristics on the occurrence of disease flare during follow-up after bDMARD/tsDMARD discontinuation. Clinical characteristics at the start of bDMARD/tsDMARD treatment and those at the start of drug discontinuation that were considered to be clinically relevant based on previous knowledge were used as predictor variables. Univariable Fine–Gray regression analyses were performed first for each predictor variable. Thereafter, all predictor variables with *p* values <0.10 in the univariable analyses were introduced into a multivariable analysis. A backward selection procedure with a cut-off significance level of 0.05 was used in the multivariable model. Risk differences are presented as adjusted hazard ratios (HRs) with 95% CIs.

For all tests, a probability value (*p* value) of <0.05 was considered to indicate statistical significance. All calculations were performed using Easy R (Saitama Medical Center, Jichi Medical University, Saitama, Japan) [25].

# Results

## Patient characteristics

As shown in Fig 1, 97 patients with RA who desired drug discontinuation after maintenance of remission or LDA for ≥48 weeks due to stable treatment with bDMARDs (46 patients with TNF inhibitors and 19 with tocilizumab) or tofacitinib (32 patients) were enrolled into this study. The participants' clinical characteristics are shown in Table 1. When treatment with these DMARDs was introduced, 44 patients (45.4%) had high CDAI values and 54 (55.7%) had early RA (≤2 years of disease duration). The mean mTSS was 16.9. The mean time from RA onset to the first use of MTX was 2.4 years, and 63 patients (64.9%) were bDMARD-naïve patients. During treatment with bDMARDs or tofacitinib, MTX was concomitantly used in 90 patients (92.8%), and 11 patients (11.3%) used prednisolone for priming when introducing bDMARDs or tofacitinib (within 3 months). Immediately before starting the bDMARD/tsDMARD discontinuation, 81 patients (83.5%) were in remission. The mean length of remission or LDA was 4.2 years. All patients received a stable dose of MTX after starting DMARD/tofacitinib discontinuation.

## Disease flare following discontinuation of bDMARDs and tofacitinib

As shown in Table 2, the mean follow-up period after discontinuation was 2.1 years (SD 2.0), and disease flare occurred in 74 patients (76.3%). Crude IRs of disease flare during follow-up were 0.36 per PY (95% CI 0.29–0.45). CIF plots for the probability of disease flare from the discontinuation of bDMARDs and tofacitinib are shown in Fig 2A. The estimated median time to disease flare was 1.6 years (95% CI 0.9–2.6). The mean of the cumulative flare probability was estimated to be 0.45 (95% CI 0.35–0.55) at 1 year, 0.56 (0.46–0.66) at 2 years, 0.64 (95% CI 0.54–0.73) at 3 years, and 0.80 (95% CI 0.68–0.87) at 5 years (Table 2). Fig 2B shows the

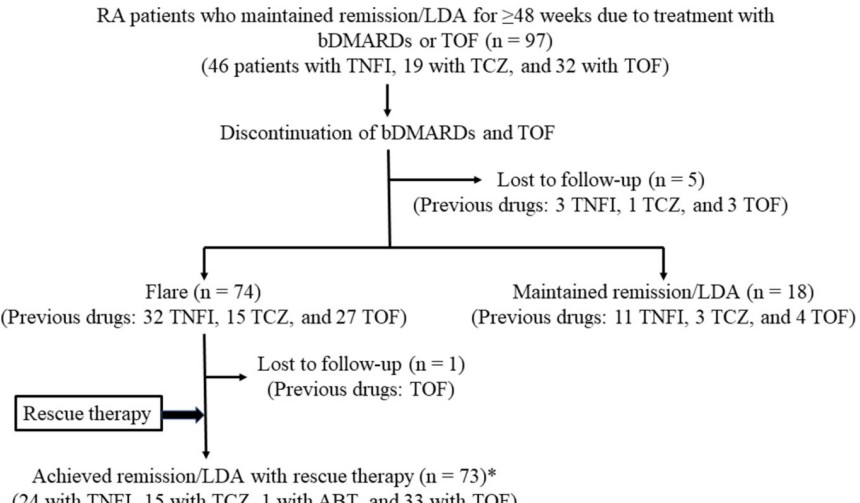

**Fig 1. Disposition of patients: Discontinuation of bDMARDs and tofacitinib and rescue therapy.** *Patients who experienced a disease flare after discontinuation of TNF inhibitors (n = 32) achieved remission or LDA with the same TNF inhibitors (n = 24), abatacept (n = 1), or tofacitinib (n = 7). The other patients regained a good outcome with previous DMARDs. RA, rheumatoid arthritis; LDA, low disease activity; bDMARDs, biological disease-modifying antirheumatic drugs; TNFI, tumor necrosis factor inhibitor; TCZ, tocilizumab; ABT, abatacept; TOF, tofacitinib.

**Table 1. Clinical characteristics of RA patients.**

| | Total (n = 97) |
|---|---|
| At the start of bDMARD or tofacitinib treatment | |
| Age, years, mean (SD) | 59.6 (11.5) |
| Male sex, number (%) | 26 (26.8) |
| RA duration, years, mean (SD) | 4.4 (7.6) |
| ≤2 years, number (%) | 54 (55.7) |
| Anti-CCP positive, number (%) | 89 (91.8) |
| RF positive, number (%) | 89 (91.8) |
| CDAI, mean (SD) | 23.8 (11.3) |
| High (CDAI >22.0), number (%) | 44 (45.4) |
| mTSS, mean (SD) | 16.9 (33.2) |
| Erosion score, mean (SD) | 7.8 (18.0) |
| No erosion, number (%) | 27 (27.8) |
| Joint space narrowing score, mean (SD) | 9.1 (15.9) |
| Time from onset to starting MTX use, years, mean (SD) | 2.4 (7.1) |
| bDMARD naïve, number (%) | 63 (64.9) |
| Treatment | |
| TNF inhibitors, number (%) | 46 (47.4) |
| Tocilizumab (IL-6 inhibitor), number (%) | 19 (19.6) |
| Tofacitinib (Janus kinase inhibitor), number (%) | 32 (33.0) |
| Concurrent use of MTX, number (%) | 90 (92.8) |
| Initial use of PSL, number (%) | 11 (11.3) |
| At drug discontinuation | |
| Age, years, mean (SD) | 63.9 (11.5) |
| RA duration, years, mean (SD) | 8.5 (8.3) |
| CDAI, mean (SD) | 1.96 (1.29) |
| Remission (CDAI ≤2.8), number (%) | 81 (83.5) |
| Length of remission or low CDAI, years, mean (SD) | 4.2 (2.9) |

RA, rheumatoid arthritis; bDMARD, biological disease-modifying antirheumatic drug; TNF, tumor necrosis factor; IL-6, interleukin-6: MTX, methotrexate; PSL, prednisolone; anti-CCP, anti-cyclic citrullinated peptide antibodies; RF, rheumatoid factor; CDAI, clinical disease activity index; mTSS, van der Heijde-modified total Sharp score; SD, standard deviation

cumulative incidence of disease flare after grouping according to the discontinued drugs (TNF inhibitors, tocilizumab, and tofacitinib). There were significant differences in flare estimates over time among these patient groups ($p$ = 0.038 for the comparison of all groups using Gray's test and $p$ = 0.023 for tofacitinib versus TNF inhibitors with the *post hoc* Holm's test). The median time to disease flare was shorter in patients who had discontinued tofacitinib compared with TNF inhibitors (0.6 years vs. 2.2 years), and the flare rate at 1 year was greater in the tofacitinib group compared with the TNF inhibitor group (66% vs. 35%) (Table 2). Kaplan–Meier plots for the flare-free survival probability after the discontinuation of bDMARDs and tofacitinib are also shown in S1 Fig.

## Rescue therapy for flares occurring after the discontinuation of bDMARDs and tofacitinib

As shown in Fig 1, all patients who experienced disease flare, except one who was lost to follow-up, started rescue therapy with the previous drugs. Among these patients, 89% recaptured

**Table 2. Flare of disease activity in RA patients after drug discontinuation.**

|  | Overall | TNF inhibitors | Tocilizumab | Tofacitinib |
|---|---|---|---|---|
|  | (n = 97) | (n = 46) | (n = 19) | (n = 32) |
| Follow-up, years, mean (SD)* | 2.1 (2.0) | 2.4 (2.2) | 2.4 (1.6) | 1.6 (2.0) |
| Follow-up, years, median (IQR)* | 1.2 (0.5–3.6) | 1.8 (0.8–4.0) | 2.6 (0.7–3.7) | 0.6 (0.3–2.2) |
| Flares, number (%) | 74 (76.3) | 32 (69.6) | 15 (78.9) | 27 (84.4) |
| Crude IR per PY (95% CI) | 0.36 (0.29–0.45) | 0.29 (0.20–0.41) | 0.33 (0.20–0.56) | 0.54 (0.37–0.79) |
| Estimates by CIF analysis |  |  |  |  |
| Time to disease flare years, median (95% CI) [†] | 1.6 (0.9–2.6) | 2.2 (1.2–4.5) | 3.1 (0.7–NA) | 0.6 (0.3–1.6) |
| Probability of disease flare, mean (95% CI) |  |  |  |  |
| At 1 year | 0.45 (0.35–0.55) | 0.35 (0.22–0.49) | 0.42 (0.20–0.63) | 0.66 (0.47–0.79) |
| At 2 years | 0.56 (0.46–0.66) | 0.50 (0.34–0.64 | 0.42 (0.20–0.63) | 0.78 (0.60–0.89) |
| At 3 years | 0.64 (0.54–0.73) | 0.57 (0.41–0.71) | 0.53 (0.29–0.72) | 0.81 (0.63–0.91) |
| At 5 years | 0.80 (0.68–0.87) | 0.77 (0.58–0.88) | – | 0.81 (0.63–0.91) |

*Follow-up was measured from the start of drug discontinuation.

[†]Median time to disease flare was the estimated time where 50% of patients would have a flare.

RA, rheumatoid arthritis; bDMARDs, biological disease-modifying antirheumatic drugs; IR, incidence rate; PY, person-year; CIF, cumulative incidence function; SD, standard deviation; IQR, interquartile range; 95% CI, 95% confidence interval

remission or LDA as follows: all patients in the tocilizumab-discontinuation group (15 patients) and the tofacitinib-discontinuation group (26 patients) regained disease control within 1 month by restarting the previous drugs; among 32 patients who discontinued TNF inhibitors, 24 (75%) successfully achieved remission within 1 month of restarting the same drug. The other eight failed to achieve LDA, but remission was reached by switching to abatacept (one patient) or tofacitinib (seven patients).

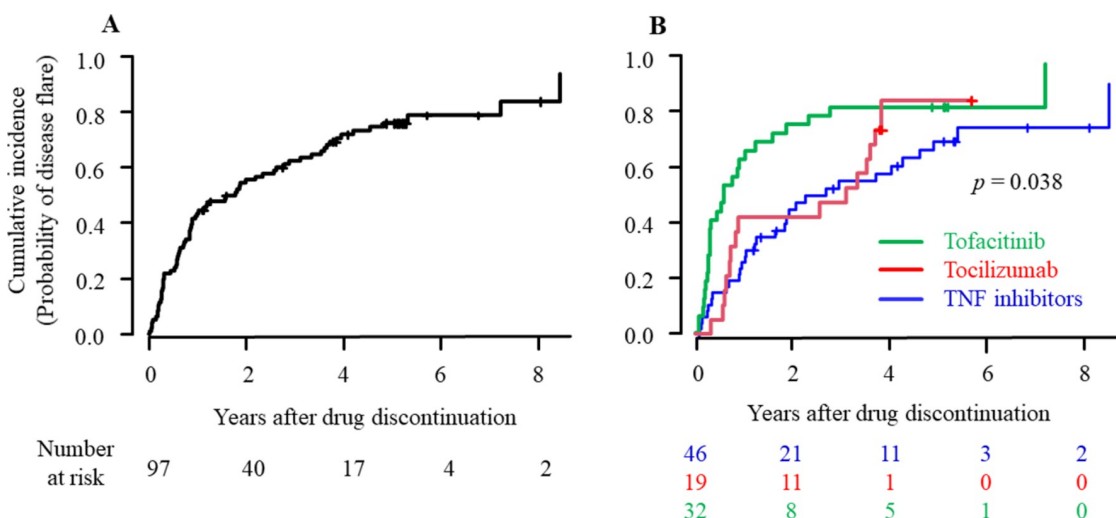

**Fig 2. Cumulative incidence of disease flare after the discontinuation of bDMARDs and tofacitinib.** CIF plots for the probability of patients who experienced disease flare following drug discontinuation are shown for all patients (A) and patients grouped according to the previous drugs (B). The cumulative flare probability over time among the patient groups were compared using Gray's test with the *post hoc* Holm's procedure. The p values are as follows: $p = 0.038$ for a comparison among the three groups; $p = 0.023$ for tofacitinib versus TNF inhibitors; $p = 0.14$ for tocilizumab versus tofacitinib; and $p = 0.53$ for TNF inhibitors versus tocilizumab. Numbers below this figure represent the number of patients remaining in the analysis. CIF, cumulative incidence function; bDMARDs, biological disease-modifying antirheumatic drugs; TNF, tumor necrosis factor.

## Influence of bDMARD/tofacitinib discontinuation on structural outcome

Thirty-one patients maintained remission for 3 years or more after drug discontinuation (previous drugs: 16 TNF inhibitors, nine tocilizumab, and six tofacitinib). Mean values of mTSS, erosion score, and joint space narrowing score (JNS) were determined at the time of starting and discontinuing bDMARD/tofacitinib treatment and 3 years after drug discontinuation. A cumulative probability plot of 3-year changes in mTSS (ΔmTSS) from drug discontinuation is shown in Fig 3. No or little radiological progression was shown in 87.1% of the patients who maintained remission for 3 years or more after drug discontinuation; the rates of patients with ΔmTSS ≤0 per 3 years and those with ΔmTSS >0 and ≤1 per 3 years were 71.0% and 16.1%, respectively.

## Predictive factors for the risk of disease flare following discontinuation of bDMARD and tofacitinib

Results of univariable and multivariable Fine–Gray competing risk regression analyses are shown in Table 3. The univariable analyses showed that RA duration, mTSS score (erosion score), and time from RA onset to first MTX use at the start of bDMARD or tofacitinib treatment, previous failure of treatment with bDMARDs, the type of treatment drugs that led to remission or LDA, and disease activity and duration of maintaining the remission or LDA status at the start of bDMARD/tofacitinib discontinuation were predictor variables with $p < 0.10$. These variables were placed into the multivariable regression analysis. In the multivariable regression model, longer RA duration, previous failure of treatment with bDMARDs, and a longer duration of remission or LDA status were identified as the predictive factors for disease flare; the adjusted HR (95% CI) was 1.06 (1.02–1.10) for RA duration per additional year ($p = 0.0041$), 0.58 (0.35–0.94) for bDMARD-naïve compared with previous bDMARD failure ($p = 0.028$), and 0.89 (0.35–0.94) for duration of remission or LDA per additional year ($p = 0.0083$). After adjusting for other predictor variables, the treatment drug type did not remain a predictive factor for disease flare.

## Safety

There were no severe adverse events that caused MTX withdrawal during the follow-up after bDMARD/tofacitinib discontinuation.

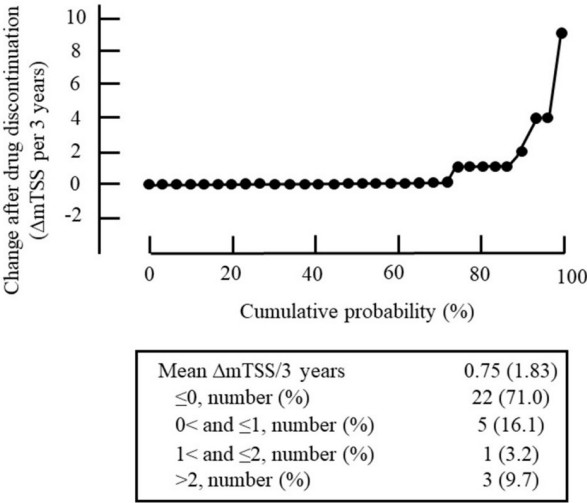

**Fig 3. Structural outcome: Cumulative probability plot of the change in mTSS due to bDMARD/tofacitinib discontinuation over 3 years.** mTSS, van der Heijde-modified total Sharp score.

**Table 3. Univariable and multivariable Fine–Gray regression analyses for disease flare after drug discontinuation.**

| Variables | Unadjusted HR (95% CI) | p | Adjusted HR (95% CI) | p |
|---|---|---|---|---|
| At the start of bDMARD or tofacitinib treatment | | | | |
| Age per additional year | 1.00 (0.98–1.02) | 0.98 | – | – |
| Male vs. female | 0.99 (0.58–1.70) | 0.97 | – | – |
| RA duration per additional year | 1.05 (1.01–1.09) | 0.026 | 1.06 (1.02–1.10) | 0.0041 |
| Anti-CCP positive vs. negative | 1.87 (0.57–6.10) | 0.30 | – | – |
| RF positive vs. negative | 1.95 (0.62–6.11) | 0.25 | – | – |
| CDAI per additional unit | 1.01 (0.99–1.02) | 0.35 | – | – |
| mTSS per additional unit | 1.00 (1.00–1.01) | 0.083 | – | – |
| Erosion score per additional unit | 1.01 (1.00–1.01) | 0.051 | – | – |
| No erosion vs. presence of erosion | 1.14 (0.67–1.92) | 0.63 | – | – |
| Joint space narrowing score per additional unit | 1.01 (1.00–1.02) | 0.19 | – | – |
| Time from onset to starting MTX use per additional year | 1.05 (1.00–1.10) | 0.051 | – | – |
| bDMARD naïve vs. previous bDMARD failure | 0.61 (0.37–0.98) | 0.043 | 0.58 (0.35–0.94) | 0.028 |
| Drug treatment | | | | |
| Drugs | | | | |
| TNF inhibitors (reference) | 1 (reference) | – | – | – |
| Tocilizumab | 1.15 (0.67–1.97) | 0.62 | – | – |
| Tofacitinib | 1.83 (1.05–3.20) | 0.034 | – | – |
| MTX use vs. no use | 1.01 (0.35–2.91) | 0.99 | – | – |
| Initial PSL use vs. no use | 0.97 (0.42–2.22) | 0.93 | – | – |
| At drug discontinuation | | | | |
| Age per additional year | 1.00 (0.98–1.02) | 0.72 | – | – |
| RA duration per additional year | 1.02 (0.98–1.07) | 0.30 | – | – |
| CDAI per additional unit | 1.02 (0.86–1.21) | 0.84 | – | – |
| Remission vs. low CDAI | 0.59 (0.31–1.10) | 0.096 | – | – |
| Length of remission or low CDAI per additional year | 0.93 (0.86–1.01) | 0.076 | 0.89 (0.82–0.97) | 0.0083 |

All predictor variables with *p* values lower than 0.10 in the univariable Fine–Gray model were introduced into multivariable analysis. For mTSS, erosion score was used. A backward stepwise selection procedure with a cut-off significant level of 0.05 was used in the multivariable model. Adjusted HRs (95% CIs) are shown for variables that remained in the final Fine–Gray model.

RA, rheumatoid arthritis; bDMARD, biological disease modifying antirheumatic drug; TNF, tumor necrosis factor; MTX, methotrexate; PSL, prednisolone; anti-CCP, anti-cyclic citrullinated peptide antibodies; RF, rheumatoid factor; CDAI, clinical disease activity index; mTSS, van der Heijde-modified total Sharp score; HR, hazard ratio; 95% CI, 95% confidence interval

## Discussion

Following bDMARD and tofacitinib discontinuation after the sustained remission or LDA status for ≥48 weeks in RA patients, disease flare occurred at an IR of 0.36 per PY, and the median time to flare was 1.6 years. No or little radiological progression was observed in 87.1% of patients who maintained remission for 3 years. In patients with a flare, 89% regained remission or LDA within 1 month by restarting the previous treatment regimens. The Fine–Gray competing risk regression analysis showed that a longer RA duration at the start of bDMARD/tofacitinib, previous failure of treatment with bDMARDs, and shorter sustained duration of remission or LDA were predictive factors for the occurrence of a disease flare after bDMARD/tofacitinib discontinuation.

In daily clinical practice, patients frequently desire drug discontinuation after reaching remission or LDA status due to the treatment with bDMARDs or tsDMARDs because of the fear of adverse events associated with these drugs. When considering drug discontinuation, one

of the most critical concerns is whether patients can rapidly recover their baseline condition by reintroducing the previous DMARDs if a disease flare occurs. In the present study, 89% of patients who experienced disease flare regained disease control using rescue therapy with the previous DMARDs. Successful recapture of remission or LDA by restarting the previous regimen has been reported in recent RCTs and observational studies involving bDMARDs and tsDMARDs [13–15, 26–37]. Rapid recovery of disease control with rescue therapy may encourage rheumatologists and RA patients to make a therapeutic decision in favor of bDMARD/tsDMARD discontinuation. Patients need to be informed about the risk of disease flares and their management strategies [38]. Additionally, patient eligibility for discontinuation should be considered. In this study, patients were scheduled to start bDMARD and tsDMARD discontinuation if they maintained a CDAI-based remission or LDA for at least 48 weeks with stable bDMARD or tsDMARD treatment and had already tapered off prednisolone.

Henaux et al. performed a systematic review and meta-analysis on structural damage, and they showed that bDMARD discontinuation was more likely to increase the risk of losing remission or LDA as well as the risk of radiological progression compared with continuing the initial bDMARD regimen [12]. In the PRESERVE study, patients who started discontinuation of etanercept had disease flares and showed radiological progression at 52 weeks more frequently compared with those receiving full- or reduced-dose etanercept [39]. In the DRESS study, radiological progression was associated with a time-weighted average of disease activity for 18 months after tapering of TNF inhibitors, although there was no association between the occurrence or number of flares and radiological progression. The findings suggested that radiological progression is caused by the long-term increase in disease activity over time after tapering [40]. These studies indicated the importance of maintaining a state of remission or LDA in patients as well as the need to regularly monitor radiological progression. It is not clear whether small increases in joint damage may become significant over time. In the present study, we showed that radiological damage did not continue to progress in approximately 90% of RA patients who maintained remission over 3 years after bDMARD/tofacitinib discontinuation.

There are few long-term outcome data regarding the bDMARD/tsDMARD-free strategy. In the HONOR study, which was conducted in Japan, 52 patients who maintained remission with adalimumab plus MTX for at least 6 months started adalimumab discontinuation. During a 5-year follow-up, adalimumab-free remission, which was determined by the 28-joint disease activity score based on erythrocyte sedimentation rate, was persistent in 48% of patients at 1 year, 31% at 3 years, and 21% at 5 years. No significant radiological changes were observed in patients who sustained LDA for 5 years, and among them, eight of nine patients (88.9%) maintained structural remission defined as ΔmTSS ≤0.5 per year [37]. In the present study, the rates of flare patients were estimated to be 45% at 1 year, 64% at 3 years, and 80% at 5 years after the discontinuation of bDMARDs and tofacitinib in RA patients. Among patients achieving bDMARD/tofacitinib-free 3-year remission, 27 of 31 (87.1%) had an mTSS ≤1 over the 3 years. Similar findings were obtained between the HONOR study and the present study.

In the present study, the predictive factors for disease flare following bDMARD/tofacitinib discontinuation were longer RA duration at the start of bDMARD/tofacitinib treatment, previous failure of biological treatment, and a shorter sustained period of the remission or LDA status before the start of drug discontinuation. Previous studies also identified several patient clinical and imaging characteristics as determinant factors for failure or success to sustain remission or LDA after the start of bDMARD tapering and discontinuation. The identified characteristics included disease activity at baseline, RA duration, timing of treatment (initial or delayed), previous use of bDMARDs, residual synovial inflammation under ultrasound examinations, and deep/long-standing remission [26, 37, 41–48]. These patient characteristics may help predict the sustained clinical benefit after bDMARD/tofacitinib tapering and discontinuation.

There are several limitations to this study. First, the sample size was relatively small, which may have made it difficult to draw a definitive conclusion about predictors for disease flare after bDMARD/tofacitinib discontinuation. To estimate the sample size, we needed to input an expected probability of flare-free survival at a time point of interest. However, we were unable to determine the expected long-term survival probability due to a lack of previous data. Second, the present study was an observational study in a real-world clinical setting, and therefore, allowing bDMARD/tofacitinib discontinuation was left to the discretion of the treating physician and the patient's preference, which may have produced some bias. However, we used clear inclusion criteria for this study. We included only patients with MTX-resistant RA who had sustained remission or LDA with stable DMARD treatment (which included the type and dose) for ≥48 weeks and had already tapered off prednisolone. Additionally, we defined a disease flare based on CDAI values that were regularly monitored at each scheduled visit. We continued stable doses of MTX for all patients, but an additional use of prednisolone or other csDMARDs was not allowed to control disease activity during follow-ups. Considering such consistency in the definition of a flare and the eligibility of participants, the present study provides useful information regarding long-term outcomes in RA patients after bDMARD/tofacitinib discontinuation.

## Conclusion

In this longitudinal observational study, it was estimated that half of RA patients had disease flare within 1.6 years after discontinuing bDMARDs and tofacitinib. The timing of bDMARD/tofacitinib start (RA duration before the treatment), previous failure of treatment with bDMARDs, and the length of remission or LDA before the discontinuation were significant predictive factors for successful discontinuation. Approximately 90% of patients who maintained bDMARD/tofacitinib-free remission for 3 years had no or little radiological progression. Restarting the original treatment regimen allowed rapid recapture of disease control within 1 month in almost all flare cases. Thus, attempting bDMARD/tofacitinib-free management may be a viable option that could be pursued after achieving stable control of disease activity in RA patients, especially those who had started the treatment at early RA stages and maintained remission or LDA for a longer time. Regular monitoring of disease activity and patient education about early signs of disease flare are crucial to prevent persistent disease flares and radiological damages during bDMARD/tofacitinib-free treatment for RA.

## Supporting information

**S1 Fig. Kaplan–Meier plots: The probability of flare-free survival after the discontinuation of bDMARDs and tofacitinib.** Kaplan–Meier plots for the proportion of patients who experienced no disease flare during follow-up are shown for patients who discontinued TNF inhibitors, tocilizumab, or tofacitinib. The flare-free probability among the patient groups was compared using the log-rank test with the *post hoc* Holm's procedure. The *p* values were as follows: *p* = 0.069 for a comparison among the three groups; *p* = 0.030 for tofacitinib versus TNF inhibitors; *p* = 0.14 for tocilizumab versus tofacitinib; and *p* = 0.52 for TNF inhibitors versus tocilizumab. Median time to disease flare (95% CI) was 2.2 years (1.0–3.4 years) for TNF inhibitors, 3.1 years (0.0–6.3 years) for tocilizumab, and 0.6 years (0.2–1.0 years) for tofacitinib. Numbers below this figure represent the number of patients remaining in the analysis. bDMARDs, biological disease-modifying antirheumatic drugs; TNF, tumor necrosis factor; CI, confidence interval.
(TIF)

## Author Contributions

**Conceptualization:** Shunsuke Mori, Yukitaka Ueki.

**Data curation:** Shunsuke Mori.

**Formal analysis:** Shunsuke Mori, Akitomo Okada, Tomohiro Koga.

**Funding acquisition:** Shunsuke Mori.

**Investigation:** Shunsuke Mori, Akitomo Okada, Tomohiro Koga.

**Methodology:** Shunsuke Mori, Akitomo Okada, Tomohiro Koga.

**Project administration:** Shunsuke Mori, Yukitaka Ueki.

**Resources:** Shunsuke Mori.

**Supervision:** Shunsuke Mori.

**Validation:** Shunsuke Mori, Akitomo Okada, Tomohiro Koga, Yukitaka Ueki.

**Visualization:** Shunsuke Mori.

**Writing – original draft:** Shunsuke Mori, Akitomo Okada, Tomohiro Koga, Yukitaka Ueki.

**Writing – review & editing:** Shunsuke Mori, Akitomo Okada, Tomohiro Koga, Yukitaka Ueki.

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
