## [Decision Letter · Decision Letter 0]

4 Apr 2022

PONE-D-22-04611Outcomes after discontinuing biological drugs and tofacitinib in patients with rheumatoid arthritis: a longitudinal cohort studyPLOS ONE

Dear Dr. Mori,

Thank you for submitting your manuscript to PLOS ONE. After careful consideration, we feel that it has merit but does not fully meet PLOS ONE’s publication criteria as it currently stands. Therefore, we invite you to submit a revised version of the manuscript that addresses the points raised during the review process.

We look forward to receiving your revised manuscript.

Kind regards,

Chengappa Kavadichanda

Academic Editor

PLOS ONE

Journal Requirements:

When submitting your revision, we need you to address these additional requirements. 1. Please ensure that your manuscript meets PLOS ONE's style requirements, including those for file naming. The PLOS ONE style templates can be found at https://journals.plos.org/plosone/s/file?id=wjVg/PLOSOne_formatting_sample_main_body.pdf and https://journals.plos.org/plosone/s/file?id=ba62/PLOSOne_formatting_sample_title_authors_affiliations.pdf
 2. Thank you for stating the following in the Competing Interests section:  "I have read the journal's policy and the authors of this manuscript have the following competing interests: S. Mori has received lecture fees from Pfizer Japan Inc., Eli Lilly Japan K.K, and Asahikasei Pharma. The other authors have declared that no conflicts of interest exist. "   We note that you received funding from a commercial source: Pfizer Japan Inc., Japan K.K and Asahikasei Pharma. Please provide an amended Competing Interests Statement that explicitly states this commercial funder, along with any other relevant declarations relating to employment, consultancy, patents, products in development, marketed products, etc.  Within this Competing Interests Statement, please confirm that this does not alter your adherence to all PLOS ONE policies on sharing data and materials by including the following statement: "This does not alter our adherence to PLOS ONE policies on sharing data and materials.” (as detailed online in our guide for authors http://journals.plos.org/plosone/s/competing-interests).  If there are restrictions on sharing of data and/or materials, please state these. Please note that we cannot proceed with consideration of your article until this information has been declared.  Please include your amended Competing Interests Statement within your cover letter. We will change the online submission form on your behalf. 3. Your ethics statement should only appear in the Methods section of your manuscript. If your ethics statement is written in any section besides the Methods, please delete it from any other section. 

Reviewers' comments:

Reviewer's Responses to Questions

**Comments to the Author**

1. Is the manuscript technically sound, and do the data support the conclusions?

Reviewer #1: Yes

Reviewer #2: Yes

2. Has the statistical analysis been performed appropriately and rigorously? 

Reviewer #1: Yes

Reviewer #2: Yes

3. Have the authors made all data underlying the findings in their manuscript fully available?

Reviewer #1: No

Reviewer #2: Yes

4. Is the manuscript presented in an intelligible fashion and written in standard English?

Reviewer #1: Yes

Reviewer #2: Yes

5. Review Comments to the Author

Reviewer #1: Major issues:

Please clarify if this was a prospective cohort.

There is little data on how dDMARDs or ts DMARDs were discontinued. Where these tapered off or stopped suddenly? Was any other csDMARD allowed except MTX?

What about loss to follow-up? How was it ensured that the patients would be coming back every month (for CDAI) for such long periods?

Abstract:

It should mention the median (IQR) time on follow-up.

Short title: Should be mention relapse after bDMARD/tofacitinib discontinuation?

Introduction:

“ Although there is currently no cure…” sounds odds. What is cure if not sustained drug-free remission?

Methods:

Radiographs were obtained at what frequency?

How was normality of data checked?

Results:

Please provide SD for mean follow-up period.

How was “severe adverse effects” screened for?

Could a Kaplan Meier curve be added with log rank tests between Tofacitinib and bDMARDs?

Reviewer #2: Summary In this observational longitudinal cohort study. The investigators have looked at the impact of discontinuation of biological DMARDs or tofacitinib on Rheumatoid arthritis patients who were in clinical remission or having low disease activity.

• As if now the pointers towards drug tapering in patients with RA are not clear but studies suggest its feasibility

• This study focuses on the variables associated with increased risk of disease flare after treatment discontinuation

• The study is highly appreciable and have been done in a more meaningful way

First Impression

The research is original as well as relevant in the clinical practice and may help practitioners in stratifying their patients with RA with relatively higher risk of relapse after treatment discontinuation and to execute a better screening for considering drug tapering or stopping once they achieve remission.

• The article has been structured well and articulated in a lucid manner.

Abstract • Explicitly written, well summarized, involves the salient findings of the study.

Following point –

1. It will be nice if median duration of follow up can be mentioned in abstract

Introduction

• The introduction is well structured and provide the correct insight to the purpose of the study as well the assumed hypothesis.

Methodology

• The methodology and the work flow described is clear and easy to follow.

Following points –

1. Line 98 page 12“First-date” could be possibly rewritten as “since the day of drug discontinuation”

2. Line 103 page 12 - “When disease activity was not controlled by the previous drugs, patients were treated with

another bDMARD or tsDMARD” what was the waiting period before starting a new drug

3. Line 112 page 13 – “For patients who were refractory to” how was this refractory state defined.

4. What was the degree of agreement between two blinded readers assessing radiographs?

Statistics

• Lucid description and appropriate methods mentioned for result analysis.

Following points-

1. How sample size was calculated for this study

Results and discussion

• Organized and well-structured.

Following points –

1. Line 173 page 15 onwards – kindly also mention absolute numbers of the patients (only percentage has been mentioned)

2. Line 184 to 186 page 15 – Mention “Table 02” somewhere since these findings are summarized in same

Conclusion

• Structured to the point and gives the summary of the primary finding.

Figures and tables

• All the figures and tables are clear and self-explanatory.

Following points –

1. Few variables mentioned as mean and standard deviation appears to have standard deviation more than mean, probably because of skewed data. Such variables may be more appropriately mentioned as median with Interquartile range or range rather than mean and standard deviation

6. PLOS authors have the option to publish the peer review history of their article (what does this mean?). If published, this will include your full peer review and any attached files.

Reviewer #1: No

Reviewer #2: No

---

## [Author Response · Author response to Decision Letter 0]

28 Apr 2022

Response to Reviewers

We are grateful to the reviewers for their valuable comments as well as their time and energy that they put into reviewing our manuscript. We have made all requested changes and added new information to the manuscript in response to their insightful comments. All alterations are highlighted in red text in the revised version of the manuscript. We are confident that the manuscript has benefited from the reviewers’ useful comments and suggestions.

Below are point-by-point replies to the reviewers’ comments.

Reply to Reviewer 1

Major issues

1. We wish to thank the reviewer for the comment regarding the study design. In this study, all participants who underwent bDMARD/tofacitinib discontinuation were prospectively followed-up until disease flare or until the end of the study. To clarify this point, we included the word “prospective” in the title and revised the title as follows: “Long-term outcomes after discontinuing biological drugs and tofacitinib in patients with rheumatoid arthritis: a prospective cohort study” (lines 2 and 3). We also added the word “prospectively” to several sentences in the abstract (line 33) and the Patients and Methods section (line 96). 

2. We appreciate the reviewer’s comment that the strategy for bDMARDs/tofacitinib discontinuation should be clarified. In the present study, we used a direct DMARD discontinuation strategy instead of a gradual discontinuation strategy for patients who desired drug discontinuation after maintaining remission or low disease activity (LDA) for ≥48 weeks through stable drug treatment with bDMARD or tofacitinib. Although recent clinical studies have suggested that stepwise tapering until complete discontinuation of DMARDs could be safely implemented with little risk of flares, the optimal strategy for gradual withdrawal for each DMARD remains unknown. To clarify this point, several sentences were added to the revised manuscript (lines 95–98). As mentioned in the Results and Discussion sections of the previous version of our manuscript, we continued a stable dose of methotrexate (MTX) for all patients during follow-up, but additional use of prednisolone or other csDMARDs was not allowed to control disease activity (lines 210, 211, and 356–358). To clarify this point, we added this information to the Patients and Methods section of the revised manuscript (lines 100–102).

3. We apologize for the error. After bDMARD/tofacitinib discontinuation, clinical disease activity was not measured every month, but, rather, at each scheduled visit (usually every 8 weeks) or at any time during follow-up if patients had clinical signs or symptoms. In our rheumatology department, all patients need to make an appointment for next medical examinations during follow-up. We recommended that the patients should return for a follow-up visit every 8 weeks after bDMARD/tofacitinib discontinuation. If patients missed at least two scheduled visits without any contact, they were considered to be lost to follow-up. For patients who experienced a disease flare after bDMARD/tofacitinib discontinuation, disease activity was examined every 2 weeks during rescue therapy. We included this information in the revised manuscript (lines 98, 99, 104, 105, 114, 115, 134–136, 355, and 356). 

Abstract

1. We wish to thank the reviewer for the comment that follow-up duration should be described in the abstract. The median (interquartile range [IQR]) length of follow-up was 1.2 years (0.5–3.6 years). However, recent recommendations for statistical analysis in rheumatology studies state that the mean and standard deviation (SD) are meaningful descriptive statistics for data following continuous distributions, even for non-normally distributed data (Recommendation #9: Lydersen S. Statistical review: frequently given comments. Ann Rheum Dis 2015; 74, 325-325; Lydersen S. Mean and standard deviation or median and quartiles? Tidsskr Nor Laegeforen 2020; 140(9), pii 20-0032). Additionally, an in-depth discussion on the “mean and SD” was published by Professor Lydersen in 2021 (Ann Rheum Dis 2021; doi:10.1136/annrheumdis-2021-220336). In the present study, in accordance with this recommendation, we used the mean and SD as descriptive statistics for all data with continuous distributions except the median time to disease flare, which was computed in the survival analysis. We added a mean follow-up of 2.1 years (SD 2.0) to the abstract (lines 35 and 36 in the abstract). This information on descriptive statistics was added to the Patients and Methods section (lines 164–166). New references were provided (refs. 21 and 22). We understand that the median (IQR) is often used to show the distribution of follow-up time data, and we also included these data in Table 2. 

Short title

1. We appreciate the reviewer’s suggestion to use in the short title “Relapse after bDMARD/tofacitinib discontinuation.” In the present study, we indicated that, besides the risk of disease flare after bDMARD/tofacitinib discontinuation, restarting the previous treatment regimens led to rapidly regaining disease control in 89% of flare patients within 1 month. Flares are manageable in most patients with rheumatoid arthritis. Considering this point and the reviewer’s suggestion, we revised the short title to “Disease flare and rescue therapy after bDMARD/tofacitinib discontinuation” (line 25).

Introduction

1. We understand the reviewer’s concern regarding the description “although there is currently no cure.” This sentence may lead to a misunderstanding by the reader. We deleted this sentence in the revised manuscript. 

Patients and Methods

1. We appreciate the reviewer’s comment regarding the assessment of joint destruction. In this study, we took radiographs of the hands and feet at the start of bDMARD or tofacitinib treatment, at the start of bDMARD/tofacitinib discontinuation, and every year during follow-up after discontinuation. A change in van der Heijde-modified total Sharp score (mTSS) between the bDMARD/tsDMARD discontinuation and 3 years later (ΔmTSS) was determined for each patient who had maintained remission for 3 years or more. The average of the mTSS scores that were determined by two readers was used for each patient’s mTSS. We added this information to the revised manuscript (lines 140–142 and 147–150).

2. We appreciate the reviewer’s comment on the normality of the mTSS data. As the reviewer pointed out, the data did not follow a normal distribution. As we mentioned in the response to Comment #1 about the abstract, however, recent recommendations for statistical analysis in rheumatology studies state that, as descriptive statistics, the mean (SD) is meaningful and well defined for data with continuous distributions, even for non-normally distributed data (refs. 21 and 22). Additionally, in recent studies, some of which were cited in this manuscript, the mean and SD were used for the mTSS/ΔmTSS data (e.g., refs 29, 33, 37, 39, and 40). Thus, we used the mean (SD) for mTSS and ΔmTSS (Table 1 and Fig. 3).

Results

1. In accordance with the reviewer’s comment, we added SD for the mean follow-up period (lines 214 and 215). The data are also shown in Table 2.

2. We appreciate the reviewer’s comment regarding screening for severe adverse events. As mentioned in the Patients and Methods section of the previous manuscript, all participants were allowed to receive a stable dose of MTX after discontinuing bDMARDs and tofacitinib during follow-up (lines 99 and 100). Severe adverse events were defined as those that caused MTX discontinuation during follow-up. Decisions to discontinue MTX due to adverse events were made by the treating physicians based on a comprehensive evaluation of physical findings, laboratory findings, and radiological examinations at every visit or at any time when patients had clinical signs or symptoms. We included this information in the Patients and Methods section of the revised manuscript (lines 106–110 and 273).

3. We appreciate the reviewer’s suggestion to add Kaplan–Meier (K–M) plots with the log-rank test. In the present study, we used the cumulative incidence function (CIF) to estimate the probability of the occurrence of a flare event over time, because we considered the presence of competing risks. As the reviewer pointed out in Major Issues Comment #3, there were lost to follow-up events in this study, which were treated as competing risk events in survival analysis. The occurrence of a competing risk event precludes the occurrence of the primary event of interest. In the presence of competing risks, the simple use of the K–M survival function can overestimate the cumulative incidence probability of a flare event. To avoid this possibility, we used the CIF instead of the K–M survival function. Gray’s test for the CIF model was used to test the equality of CIF plots among patient groups. Gray’s test is the analogue to the log-rank test that is used to test the equality of K–M survival curves between groups. To clarify this point, we added this information to the Patients and Methods section of the revised manuscripts (lines 169–180). New references were also provided (refs. 23 and 24). We agree with the reviewer’s suggestion that it may be useful to provide both CIF plots and K–M plots. Thus, we added K–M plots for the flare-free survival probability after bDMARD/tofacitinib discontinuation in the revised manuscript (lines 228–230, supporting information lines 378–392 and Fig S1). 

Reply to Reviewer 2

We appreciate the reviewer’s comments about our study as a whole.

Abstract

1. We thank the reviewer for the suggestion to include the median duration of follow-up in the abstract. The median duration of follow-up (interquartile range [IQR]) was 1.2 years (0.5–3.6 years). However, recent recommendations for statistical analysis in rheumatology studies state that the mean and standard deviation (SD) are meaningful and well defined for data following all types of continuous distributions, even for non-normally distributed data (Lydersen S. Statistical review: frequently given comments. Ann Rheum Dis 2015; 74, 325-325; Lydersen S. Mean and standard deviation or median and quartiles? Tidsskr Nor Laegeforen 2020; 140, pii: 20-0032). We therefore included the mean follow-up (SD) in the abstract of the revised manuscript (lines 35 and 36). We also added an explanation that the mean and SD were used as descriptive statistics for data with continuous distributions, even for non-normally distributed data in the Patients and Methods section (lines 164–166). New references were provided (refs. 21 and 22). We understand that median (IQR) is often used to show the distribution of follow-up time data, and we also included these data in Table 2.

Methodology

1. In accordance with the reviewer’s comment, we revised this sentence to “Follow-up started on the day of bDMARD/tsDMARD discontinuation” (lines 102 and 103).

2. We appreciate the reviewer’s comment regarding the waiting period before the start of a new DMARD. Patients who experienced a disease flare started rescue therapy with previous DMARDs. Disease activity was examined every 2 weeks during rescue therapy. We changed previous bDMARDs or tsDMARDs to another drug if disease activity was not controlled by the previous drugs by 4 weeks. We added this information to the Patients and Methods section (lines 114–116).

3. We appreciate the review’s comment on the definition of “refractory to the original subcutaneous regimen of tocilizumab.” For patients who failed to achieve remission or low disease activity (LDA) by the original subcutaneous regimen at 3 months, a 162-mg once-weekly regimen was used. We added this information to the revised manuscript (lines 125–127).

4. We thank the review for the comment regarding the degree of agreement between two blinded readers who scored the radiographs of the hands and feet. These two readers (Drs. Okada and Koga) completed a training course on the Sharp-van der Heijde method at Leiden University Medical Center and received a Certificate of Completion that was issued by Professor van der Heijde. In this study, they independently scored each radiograph using the van der Heijde-modified total Sharp score (mTSS). To ensure objectivity of the scoring, the average of the mTSS scores determined by each of the two readers was used for each patient’s mTSS. We included this information in the Patients and Methods section of the revised manuscript (lines 147 and 148). 

Statistics

1. We appreciate the reviewer’s comment on sample size estimation. To estimate the sample size in a survival study for time-to-event data, we needed to input the expected probability of event-free survival at a time point of interest. The present study was a prospective follow-up to evaluate long-term outcomes of bDMARD/tofacitinib discontinuation. We estimated an appropriate probability of disease flare/probability of flare-free survival after direct bDMARD/tofacitinib discontinuation based on previous data on the long-term survival probability. As mentioned in the Introduction section, however, follow-up periods after discontinuation of drugs were within 1 year in most studies (lines 73 and 74). There are few long-term outcome data regarding the bDMARD/tsDMARD-free strategy after remission or LDA achievement. In the HONOR study, which was an open-label non-randomized study, 52 patients who maintained remission with adalimumab plus MTX for at least 6 months started adalimumab discontinuation. During a 5-year follow-up, adalimumab-free remission was persistent in 21% of patients at 5 years (a flare-free survival probability of 0.21 at 5 years) (ref. 37, lines 319–325). However, the inclusion criteria (the status of disease activity and its duration before starting adalimumab discontinuation) were different from ours. Additionally, long-term survival data regarding other tumor necrosis factor inhibitors, tocilizumab, or tofacitinib (e.g., flare-free survival probability at 5 years after drug discontinuation) were not available in the literature. It was difficult to estimate a long-term probability of disease flare or flare-free survival due to a lack of previous data. We were therefore unable to estimate the sample size for this study. We hope our long-term survival data will be helpful to calculate sample size for future studies. We added an explanation that we could not estimate the sample size for this study due to the lack of previous data in the Discussion section of the revised manuscript (lines 346–349).

Results and Discussion

1. We agree with the reviewer’s suggestion to include the absolute patient number in this section. Thus, we changed “percentage” to “patient number (percentage)” in the revised version of the text. These data are also shown in Table 1 (lines 201–209).

2. In accordance with the reviewer’s comment, we included “As shown in Table 2” at the beginning of this sentence in the revised manuscript (line 214).

Figures and Tables

1. We understand the reviewer’s concern regarding the use of the mean and SD for a skewed data set. As we mentioned in the response to Comment #1 about the abstract, however, recent recommendations for statistical analysis in rheumatology studies state that as descriptive statistics, the mean and SD are well defined and relevant for data with continuous distributions, even for non-normally distributed data (refs. 21 and 22). Additionally, an in-depth discussion on the “mean and SD” was published by Professor Lydersen in 2021 (Ann Rheum Dis 2021; doi:10.1136/annrheumdis-2021-220336). Recent rheumatology studies, some of which are presented in this study, used the mean and SD for baseline patient characteristics even though they were skewed distributed data. Considering these points, we used the mean and SD as descriptive statistics for all data with continuous distributions except the median time to disease flare, which was computed in the survival analysis. As mentioned above, we added this information to the Patients and Methods section (lines 164–166).

---

## [Decision Letter · Decision Letter 1]

10 Jun 2022

Long-term outcomes after discontinuing biological drugs and tofacitinib in patients with rheumatoid arthritis: a prospective cohort study

PONE-D-22-04611R1

Dear Dr. Mori,

We’re pleased to inform you that your manuscript has been judged scientifically suitable for publication and will be formally accepted for publication once it meets all outstanding technical requirements.

Kind regards,

Chengappa Kavadichanda

Academic Editor

PLOS ONE

Additional Editor Comments (optional):

Reviewers' comments:

Reviewer's Responses to Questions

**Comments to the Author**

1. If the authors have adequately addressed your comments raised in a previous round of review and you feel that this manuscript is now acceptable for publication, you may indicate that here to bypass the “Comments to the Author” section, enter your conflict of interest statement in the “Confidential to Editor” section, and submit your "Accept" recommendation.

Reviewer #1: All comments have been addressed

Reviewer #2: All comments have been addressed

2. Is the manuscript technically sound, and do the data support the conclusions?

Reviewer #1: Yes

Reviewer #2: Yes

3. Has the statistical analysis been performed appropriately and rigorously? 

Reviewer #1: Yes

Reviewer #2: Yes

4. Have the authors made all data underlying the findings in their manuscript fully available?

Reviewer #1: Yes

Reviewer #2: Yes

5. Is the manuscript presented in an intelligible fashion and written in standard English?

Reviewer #1: Yes

Reviewer #2: Yes

6. Review Comments to the Author

Reviewer #1: Thank you for the clarifications and corrections.

Thank you for the clarifications and corrections.

Thank you for the clarifications and corrections.

Thank you for the clarifications and corrections.

Thank you for the clarifications and corrections.

Thank you for the clarifications and corrections.

Reviewer #2: Thank you for the adequate response to all the comments. All comments have been addressed adequately

7. PLOS authors have the option to publish the peer review history of their article (what does this mean?). If published, this will include your full peer review and any attached files.

Reviewer #1: No

Reviewer #2: No

---

## [Editor Report · Acceptance letter]

13 Jun 2022

PONE-D-22-04611R1 

Long-term outcomes after discontinuing biological drugs and tofacitinib in patients with rheumatoid arthritis: a prospective cohort study 

Dear Dr. Mori:

I'm pleased to inform you that your manuscript has been deemed suitable for publication in PLOS ONE. Congratulations! Your manuscript is now with our production department. 

Kind regards, 

on behalf of

Dr. Chengappa Kavadichanda 

Academic Editor

PLOS ONE